# Quantifying and Mitigating Hospital Domain Bias in Pathology Foundation Models using Adversarial Feature Disentanglement

**Mengliang Zhang**[1]                                    MXZ3935@MAVS.UTA.EDU

[1] *CSE Department, The University of Texas at Arlington*

## Abstract

Pathology foundation models (PFMs) have demonstrated remarkable potential in whole-slide image (WSI) diagnosis. However, pathology images from different hospitals exhibit domain shifts due to variations in scanning hardware and preprocessing. These differences cause PFMs to learn spurious hospital-specific features, severely compromising their robustness and generalizability in clinical settings. We present the first systematic study of this hospital-source domain bias in PFMs. To address the critical trade-off between diagnostic utility and domain predictability, we establish a quantification pipeline and introduce the Robustness Index (RI). Furthermore, we propose a lightweight adversarial framework for feature disentanglement. This framework employs a trainable adapter and a domain classifier connected via a Gradient Reversal Layer (GRL) to remove latent hospital-specific information from frozen PFM representations without modifying the encoder itself. Experiments on multi-center histopathology datasets demonstrate that our approach substantially suppresses domain predictability and achieves significant gains in feature robustness. Crucially, the method maintains or improves disease classification performance, proving its efficacy particularly in out-of-domain scenarios. Our code is provided at: https://github.com/MengRes/pfm_domain_bias

**Keywords:** Pathology Image, Domain Bias, Foundation model.

## 1. Introduction

Whole-slide imaging (WSI) has become a key tool in digital pathology, enabling scalable analysis of gigapixel histopathology slides. In most downstream applications, WSIs are partitioned into small patches, and deep models—such as ResNet (He et al., 2016), Vision Transformer (ViT) (Dosovitskiy et al., 2021), or foundation encoders like UNI (Chen et al., 2024), PLIP (Huang et al., 2023) are employed to extract visual features for tasks including disease classification, tumor grading, and subtype analysis. These patch-level features form the backbone of many computational pathology pipelines.

However, a critical yet often underexplored challenge in this setting is the presence of domain-specific bias in the data. Patches collected from different hospitals or scanners frequently differ in staining protocols, image resolution, scanner artifacts, and tissue preparation. Such variations introduce spurious correlations into the learned representations, causing models to inadvertently rely on hospital-specific cues rather than true disease-related signals, severely compromising their robustness and generalizability in clinical settings. Even state-of-the-art pathology foundation models (PFMs), such as Phikon (Filiot

et al., 2023) and UNI (Chen et al., 2024), exhibit a significant degree of domain bias. Visual confirmation of this phenomenon is provided in Figure 3, where t-SNE (Van der Maaten and Hinton, 2008) clustering of patch embeddings from the TCGA-BRCA (The Cancer Genome Atlas Network, 2012) dataset clearly shows that patches from the same hospital source cluster tightly together. This indicates the learned features inadvertently retain substantial hospital-specific information.

Several prior studies have explored domain bias in PFMs. Vaidya et al. (Vaidya et al., 2024) examined performance disparities across patient populations, while Lin et al. investigated inter-hospital variations. de Jong et al. (de Jong et al., 2025) proposed a robustness metric using a KNN-based approach. Other studies attempted to mitigate domain shifts through non-learnable stain normalization techniques, such as Macenko (M. Macenko et al., 2009) normalization, which only address low-level color differences. Another line of work leverages parameter-efficient fine-tuning methods, such as LoRA (Low-Rank Adaptation) (Hu et al., 2021), to adapt PFMs to out-of-distribution (OOD) data. Yet, these approaches have notable limitations: the KNN-based metric lacks a comprehensive dataset-level assessment; stain normalization fails to resolve complex, non-stain-related domain shifts; and LoRA fine-tuning approaches do not explicitly ensure the reduction of domain bias while preserving diagnostic performance. Thus, there is a clear need for a systematic, parameter-efficient framework that can rigorously quantify and explicitly mitigate domain bias while guaranteeing the preservation of diagnostic performance.

In this work, a systematic pipeline is established to evaluate PFM domain bias, using separate Multi-Layer Perceptron (MLP) training to quantify severity via hospital classification AUC. We introduce the Robustness Index to quantify the utility-predictability trade-off. To mitigate bias, we propose a lightweight adversarial framework utilizing a trainable projection head, disease classifier, and a domain classifier connected by a Gradient Reversal Layer (GRL). During backpropagation, the GRL reverses the gradient, suppressing hospital-related cues while preserving disease information. Experimental results show the framework effectively suppresses hospital-specific signals, alleviating domain bias, and achieving substantial robustness gains on multi-center datasets

Our main contributions are as follows: 1. We establish a systematic pipeline to identify and quantify hospital-source domain bias in PFM features, introducing the Robustness Index (RI) to assess the net utility of the learned representations. 2. We propose a lightweight adversarial training framework that incorporates a Gradient Reversal Layer (GRL) to suppress hospital-related discriminative cues in image features without modifying the base PFM encoder, ensuring parameter efficiency and core feature preservation. 3. Experiments on multi-center histopathology datasets demonstrate that our method significantly suppresses latent hospital information while maintaining or enhancing disease classification performance, achieving substantial gains in robustness (quantified by $\Delta RI$).

## 2. Related Works

### 2.1. Pathology Foundation Models (PFMs)

The initial success of general vision models like CLIP (Radford et al., 2021) and DINO (Caron et al., 2021) in pathology paved the way for models specifically optimized for the domain. More recently, several dedicated Pathology Foundation Models (PFMs) have been

proposed, including UNI (Chen et al., 2024), CONCH (Lu et al., 2024), GIGA_PATH (Xu et al., 2024), and VIRCHOW (Vorontsov et al., 2024). These models, typically pretrained on large-scale natural image corpora or multimodal datasets, have demonstrated impressive zero-shot and few-shot capabilities in various pathology tasks, such as tumor classification, subtyping, and grading. They are commonly employed as frozen encoders to extract highly transferable, high-dimensional features from image patches. Despite their strong semantic capabilities, studies consistently observe that the features extracted by these models still encode significant dataset- or site-specific biases, including scanner artifacts, staining variations, and patient demographics.

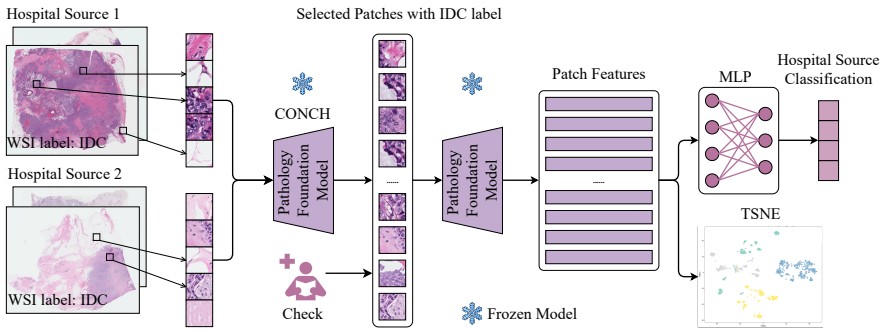

Figure 1: Pipeline for evaluating hospital-domain bias in pathology foundation models. Only patches consistent with their WSI labels are used. A simple multi-layer perceptron (MLP) is trained to classify hospital sources, where higher accuracy indicates stronger domain bias.

## 2.2. Domain Bias Mitigation and Disentanglement

Prior work has approached hospital-domain bias from various angles. Lin et al. (Lin et al., 2025) systematically unveiled institution-specific biases in pathology foundation models, analyzing causes ranging from scanner hardware to staining protocols. Similarly, Kheiri et al. (Kheiri et al., 2025) conducted a comprehensive investigation into potential bias factors within histopathology datasets, auditing demographic and site-specific confounders. While these works provide crucial diagnostic insights and survey potential solutions, they primarily focus on investigating the existence of bias rather than proposing a specialized algorithmic mitigation pipeline for frozen foundation models.

To address these shifts, early methods focused on pixel-space preprocessing, such as stain normalization (M. Macenko et al., 2009; A. Vahadane et al., 2016), to alleviate color variations. However, these fail to address complex, non-stain-related domain shifts (e.g., scanner artifacts). In the feature space, Bidgoli et al. (Asilian Bidgoli et al., 2022) proposed a deep feature selection method to discard features highly correlated with bias. However, such subtractive approaches risk losing diagnostic information entangled with domain artifacts. Other works focus on adaptation: Vaidya et al. (Vaidya et al., 2024) introduced LoRA-based fine-tuning to adapt PFMs to out-of-distribution (OOD) data, and Edwin et al. (de Jong et al., 2025) proposed a KNN-based metric to measure bias correlation.

Distinct from these approaches, our framework leverages adversarial feature learning via the Gradient Reversal Layer (GRL) (Ganin et al., 2017). Unlike (Asilian Bidgoli et al., 2022) which selects features, our method actively disentangles them by learning a non-linear projection that suppresses hospital-specific cues while preserving disease patterns. Furthermore, unlike adaptation methods that require fine-tuning (Vaidya et al., 2024), our approach provides a unified, lightweight, and PFM-agnostic solution specifically designed for the frozen feature space of modern PFMs.

## 3. Methodology

### 3.1. WSI Cohort Construction and Patch Extraction

For a given WSI collection, we first construct a clinical cohort by filtering WSIs based on available demographic information (e.g., sex and age) to minimize the potential confounding effects of demographic factors on WSI disease features. To ensure fair assessment of domain shift, we select a fixed number of WSIs per disease category and hospital source, striving to maintain balanced domain and task distributions.

For each WSI, we follow a standard procedure for feature extraction. We perform tissue segmentation using Otsu thresholding to automatically generate a tissue mask, consistent with the strategy used in CLAM (Lu et al., 2021). Within the identified tissue regions, fixed-size patches are extracted via a sliding window with a predefined grid pattern. Each patch subsequently undergoes quality control, including checks for minimum effective tissue area, ensuring the retained patches possess sufficient diagnostic value. Process details are provided in Appendix A.

### 3.2. Feature Filtering for High-Fidelity Patches

The extracted patch set is further refined to ensure consistency between patch-level content and WSI-level labels. This step is critical because WSI labels are case-level, and randomly sampled patches may not always reflect the WSI-level diagnosis (e.g., containing normal tissue). We leverage the CONCH (Lu et al., 2024) model, primarily motivated by its strong zero-shot classification capabilities to accurately verify patch content and ensure high-fidelity labeling. We perform zero-shot patch classification using CONCH to obtain the top-1 predicted label and its probability. We retain only those patches where (1) the predicted label probability exceeds a predefined threshold (e.g., 0.8) and (2) the patch-level label matches the WSI-level ground truth. The high confidence threshold is chosen to maximize the reliability of the patch-level content. A subsequent manual inspection of sampled patches is additionally performed to ensure label accuracy and consistency. The process pipeline can be seen in Figure 1.

### 3.3. Domain Bias Quantification Baseline

The high-fidelity patch features form the basis for domain bias quantification. We extract features $f_i = E(x_i)$ from the frozen PFM encoder $E(\cdot)$. We adhere to this frozen setting to reflect the standard "off-the-shelf" usage of PFMs in resource-constrained clinical environments, ensuring our evaluation captures the intrinsic bias of the pre-trained models without the prohibitive cost of full-parameter fine-tuning.

To measure latent information, we train separate Multi-Layer Perceptrons (MLPs) for hospital-source classification ($\mathcal{C}_d$) and disease classification ($\mathcal{C}_y$). MLPs are employed rather than simple linear probes to capture complex, non-linear correlations between the embeddings and domain artifacts. Both MLPs consist of two hidden layers with ReLU activations, taking the PFM feature dimension $D$ as input. Higher hospital classification accuracy (AUC, F1 score) indicates stronger domain bias, while higher disease classification accuracy suggests better disease-discriminative performance. To systematically quantify the trade-off between bias mitigation and diagnostic utility, we introduce the **Robustness Index (RI)**. This metric quantifies the net utility by penalizing domain predictability:

$$RI = \mathcal{A}_{Disease} - (\mathcal{A}_{Hosp} - \mathcal{A}_{Hosp,Random}) \tag{1}$$

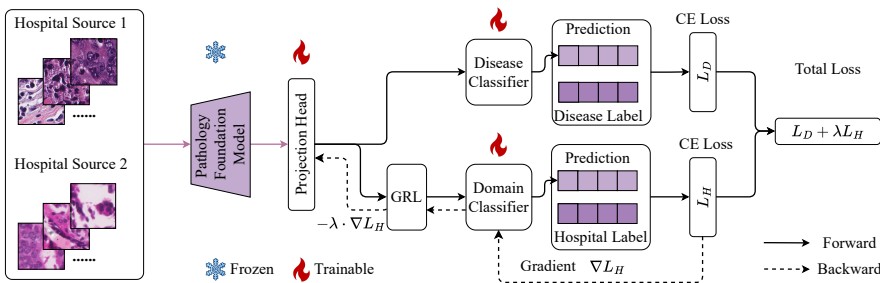

Figure 2: Advarsarial training framework.

where $\mathcal{A}_{\text{Hosp, Random}}$ represents the performance of a random classifier, which is set to 0.5 for the multi-class AUC baseline. We choose AUC for both $\mathcal{A}_{\text{Disease}}$ and $\mathcal{A}_{\text{Hosp}}$ due to its robustness against class imbalance. A higher RI signifies a feature set with both high diagnostic utility and low domain predictability. This index will be used to calculate the Robustness Improvement Index ($\Delta$RI) in our results section to compare the net gain of our adversarial framework against the baseline MLP.

### 3.4. Adversarial Disentanglement Framework

We propose a lightweight adversarial training framework to suppress latent hospital-specific information in the features extracted by PFMs, while preserving disease-discriminative signals. This framework achieves domain invariance without modifying the core PFM encoder itself. As illustrated in Figure 2, the framework incorporates three trainable components: a projection head ($A$), a domain classifier ($\mathcal{C}_d$), and a disease classifier ($\mathcal{C}_y$). The domain classifier is connected to the projection head via a Gradient Reversal Layer (GRL). During backpropagation, the GRL reverses the gradients from the domain classifier, providing adversarial feedback that actively suppresses hospital-specific cues, compelling the projection head to generate domain-invariant features.

**Problem Setup.** Let $\mathcal{D} = \{(x_i, y_i, d_i)\}_{i=1}^{N}$ denote a dataset of WSI patches, where $x_i \in \mathbb{R}^{H \times W \times 3}$ is an image patch, $y_i$ is the disease label (task label), and $d_i$ is the domain label indicating the source hospital. We assume access to a frozen encoder $E(\cdot)$ (e.g., UNI or

CONCH), which maps the patch $x_i$ to a feature vector:

$$f_i = E(x_i) \in \mathbb{R}^D \tag{2}$$

Our objective is to learn a transformed, low-dimensional representation $z_i$ through the trainable projection head $A(\cdot)$, such that the following two conditions are met: Utility Preservation, where $z_i$ retains sufficient disease-discriminative information for reliable prediction of $y_i$, and Bias Suppression, where $z_i$ suppresses hospital-specific information, preventing reliable prediction of $d_i$.

**Model Components.** The model comprises four components:

1. Frozen Encoder $E(x)$: Extracts patch-level features; parameters remain frozen during training.

2. Projection Head $A(f)$: A lightweight MLP that projects $f_i$ into $z_i \in \mathbb{R}^{D'}$:

$$z_i = A(E(x_i)) \in \mathbb{R}^{D'} \tag{3}$$

3. Disease Classifier $C_y(z)$: Predicts the disease label $\hat{y}_i = C_y(z_i)$.

4. Domain Classifier $C_d(\cdot)$ with GRL: Predicts the hospital source $\hat{d}_i$ after passing $z_i$ through a GRL:

$$\hat{d}_i = C_d\big(\mathrm{GRL}(z_i)\big) \tag{4}$$

As illustrated in Figure 2, the GRL acts as the identity function in the forward pass, $GRL(z_i) = z_i$. However, it reverses and scales the gradients in the backward pass:

$$\frac{\partial \mathcal{L}}{\partial z_i} \leftarrow -\lambda \cdot \frac{\partial \mathcal{L}}{\partial z_i} \tag{5}$$

where $\lambda \geq 0$ controls the adversarial strength. This mechanism forces the trainable projection head $A(\cdot)$ to produce features $z_i$ that remain informative for disease prediction while simultaneously becoming uninformative for hospital-source classification.

**Objective Function.** The total loss is a weighted sum of the disease classification loss ($\mathcal{L}_{\mathrm{D}}$) and the domain classification loss ($\mathcal{L}_{\mathrm{H}}$):

$$\mathcal{L}_{\mathrm{total}} = \mathcal{L}_{\mathrm{D}} + \lambda \cdot \mathcal{L}_{\mathrm{H}}, \tag{6}$$

where the weighting factor $\lambda$ is the same parameter used by the GRL scaling factor.

For disease Loss, we employ standard cross-entropy (CE) loss for the supervised disease prediction task:

$$\mathcal{L}_{\mathrm{D}} = \frac{1}{N} \sum_{i=1}^{N} \mathrm{CE}\big(C_y(\mathbf{z}_i), y_i\big). \tag{7}$$

For domain loss, the domain classifier $\mathcal{C}_d$ is trained to predict the hospital source using cross-entropy loss:

$$\mathcal{L}_{\mathrm{H}} = \frac{1}{N} \sum_{i=1}^{N} \mathrm{CE}\big(C_d(\mathrm{GRL}(\mathbf{z}_i)), d_i\big). \tag{8}$$

During training, only the projection head $A(\cdot)$, the disease classifier $\mathcal{C}_y$, and the domain classifier $\mathcal{C}_d$ are updated. The PFM encoder $E(\cdot)$ remains frozen. At inference time, the GRL and the domain classifier are discarded, and the final prediction is obtained solely from the disease branch:

$$\hat{y}_i = C_y\big(A(E(x_i))\big) = C_y(\mathbf{z}_i). \tag{9}$$

## 4. Experiment

### 4.1. Setup

As an illustrative example, we focus on whole-slide image (WSI) samples from the TCGA-BRCA dataset (The Cancer Genome Atlas Network, 2012)(We also use other dataset in experiments, see Section 4.3 and Appendix Section A). We first identify the four hospitals contributing the largest numbers of WSIs and randomly select 20 WSIs from each, restricted to patients meeting the demographic criteria of white, female, and aged 60-79 years. From each selected WSI, we uniformly sample 500 image patches of size $256 \times 256$ at a 40x magnification. These initial patches are subsequently filtered using the CONCH model: only patches whose zero-shot predicted label matches the WSI-level ground truth with a confidence score of at least 0.8 are retained. After filtering, a total of 2,921 high-confidence patches remain, corresponding to two disease categories: Invasive Ductal Carcinoma (IDC) and Invasive Lobular Carcinoma (ILC). The detailed statistics of this filtered cohort are presented in Appendix Table 4.

The filtered patches are then used to extract features from a set of representative pathology foundation models, including ResNet-50 (He et al., 2016), Giga-Path (Xu et al., 2024), UNI (Chen et al., 2024), UNI2-H (Chen et al., 2024), CONCH (Lu et al., 2024), TITAN (Ding et al., 2025), MUSK (Xiang et al., 2025), H-Optimus-0 (Saillard et al., 2024), Phikon (Filiot et al., 2023), Phikon-v2 (Filiot et al., 2024), and Virchow (Vorontsov et al., 2024). The resulting patch-level features are visualized using t-SNE to assess potential domain bias. For model training, we use 30 epochs with a batch size of 64, set the adversarial strength $\lambda$ to 1.0, and use a projection head with a hidden dimension of 512. Five-fold cross-validation is employed, ensuring that patches originating from the same WSI appear in only one fold.

### 4.2. Evaluation Metrics

To provide a comprehensive measure of our framework's success in achieving the two conflicting goals, we use the Robustness Improvement Index ($\Delta RI$). The base Robustness Index (RI), which is formalized in Section 3.3, quantifies the margin between the feature's task-discriminative utility ($\mathcal{A}_{Disease}$) and its inherent domain predictability ($\mathcal{A}_{Hosp}$). We adopt AUC for both $\mathcal{A}_{Disease}$ and $\mathcal{A}_{Hosp}$ due to its robustness against class imbalance. The final $\Delta RI$ quantifies the net gain achieved by our adversarial framework compared to the initial frozen feature baseline (MLP):

$$\Delta RI = RI_{Adversarial} - RI_{MLP} \tag{10}$$

A positive $\Delta RI$ signifies a successful disentanglement: the reduction in domain predictability ($\mathcal{A}_{Hosp}$) outweighs any potential loss in diagnostic power ($\mathcal{A}_{Disease}$). We also report the individual scores ($\mathcal{A}_{Disease}, \mathcal{A}_{Hosp}$) for all models to provide performance comparisons.

Table 1: Comprehensive Performance Metrics on TCGA-BRCA: Comparison of MLP (Baseline) and Adversarial (Disentangled) Features for Disease and Hospital Classification Tasks. Mean ± Standard Deviation is reported. RI results are based on AUC values, focusing on the feature disentanglement efficacy.

| Model | Method | Disease Clsf. ($\mathcal{A}_{\mathrm{Disease}}$) | | Hospital Clsf. ($\mathcal{A}_{\mathrm{Hosp}}$) | | RI (± std) | $\Delta$RI |
| | | Accuracy | AUC | Accuracy | AUC | | |
|---|---|---|---|---|---|---|---|
| CONCH | MLP | $0.9996 \pm 0.0009$ | $1.0000 \pm 0.0000$ | $0.5633 \pm 0.0827$ | $0.8122 \pm 0.0619$ | $0.6558 \pm 0.0299$ | — |
| | Adversarial | $1.0000 \pm 0.0000$ | $1.0000 \pm 0.0000$ | $0.2250 \pm 0.1675$ | $0.5900 \pm 0.1309$ | $0.9100 \pm 0.1309$ | **0.2542** |
| GIGA PATH | MLP | $0.9021 \pm 0.0764$ | $0.9660 \pm 0.0495$ | $0.7212 \pm 0.0789$ | $0.9088 \pm 0.0737$ | $0.5642 \pm 0.0388$ | — |
| | Adversarial | $0.9227 \pm 0.0220$ | $0.9611 \pm 0.0120$ | $0.2427 \pm 0.1888$ | $0.5719 \pm 0.1104$ | $0.8892 \pm 0.1104$ | **0.3250** |
| H_OPTIMUS | MLP | $0.9062 \pm 0.0536$ | $0.9731 \pm 0.0229$ | $0.8064 \pm 0.0875$ | $0.9652 \pm 0.0167$ | $0.5079 \pm 0.0167$ | — |
| | Adversarial | $0.9173 \pm 0.0418$ | $0.9546 \pm 0.0255$ | $0.3384 \pm 0.1570$ | $0.6736 \pm 0.1333$ | $0.7810 \pm 0.1333$ | **0.2731** |
| MUSK | MLP | $0.9134 \pm 0.0289$ | $0.9770 \pm 0.0177$ | $0.6848 \pm 0.1429$ | $0.8970 \pm 0.0589$ | $0.5897 \pm 0.0448$ | — |
| | Adversarial | $0.9398 \pm 0.0127$ | $0.9748 \pm 0.0073$ | $0.2883 \pm 0.1620$ | $0.5000 \pm 0.0000$ | $0.9748 \pm 0.0000$ | **0.3851** |
| PHIKON | MLP | $0.8529 \pm 0.1096$ | $0.8906 \pm 0.1274$ | $0.8505 \pm 0.1680$ | $0.9371 \pm 0.0982$ | $0.4607 \pm 0.0105$ | — |
| | Adversarial | $0.8763 \pm 0.0921$ | $0.9259 \pm 0.0579$ | $0.4533 \pm 0.1168$ | $0.7349 \pm 0.1062$ | $0.6910 \pm 0.1062$ | **0.2303** |
| PHIKON-V2 | MLP | $0.8196 \pm 0.1532$ | $0.8792 \pm 0.1927$ | $0.8474 \pm 0.1532$ | $0.9578 \pm 0.0501$ | $0.4552 \pm 0.0036$ | — |
| | Adversarial | $0.8596 \pm 0.0856$ | $0.9365 \pm 0.0414$ | $0.4760 \pm 0.1861$ | $0.7629 \pm 0.1707$ | $0.6736 \pm 0.1707$ | **0.2184** |
| RESNET50 | MLP | $0.8124 \pm 0.0377$ | $0.9015 \pm 0.0330$ | $0.5895 \pm 0.1031$ | $0.8298 \pm 0.0700$ | $0.5902 \pm 0.0537$ | — |
| | Adversarial | $0.8580 \pm 0.0472$ | $0.9312 \pm 0.0327$ | $0.2544 \pm 0.1611$ | $0.5124 \pm 0.0248$ | $0.9188 \pm 0.0248$ | **0.3286** |
| TITAN | MLP | $0.9219 \pm 0.0472$ | $0.9773 \pm 0.0224$ | $0.6267 \pm 0.1211$ | $0.8568 \pm 0.0297$ | $0.6605 \pm 0.0395$ | — |
| | Adversarial | $0.9294 \pm 0.0155$ | $0.9817 \pm 0.0079$ | $0.2289 \pm 0.1124$ | $0.5446 \pm 0.0891$ | $0.9371 \pm 0.0891$ | **0.2766** |
| UNI | MLP | $0.9132 \pm 0.0578$ | $0.9713 \pm 0.0425$ | $0.7823 \pm 0.1443$ | $0.9341 \pm 0.0690$ | $0.5339 \pm 0.0317$ | — |
| | Adversarial | $0.9278 \pm 0.0368$ | $0.9613 \pm 0.0363$ | $0.2166 \pm 0.1130$ | $0.5788 \pm 0.0630$ | $0.8825 \pm 0.0630$ | **0.3486** |
| UNI2-H | MLP | $0.9219 \pm 0.0360$ | $0.9823 \pm 0.0083$ | $0.7687 \pm 0.1533$ | $0.9456 \pm 0.0460$ | $0.5252 \pm 0.0199$ | — |
| | Adversarial | $0.9187 \pm 0.0432$ | $0.9700 \pm 0.0199$ | $0.1928 \pm 0.1079$ | $0.5142 \pm 0.0497$ | $0.9558 \pm 0.0497$ | **0.4306** |
| VIRCHOW | MLP | $0.9100 \pm 0.0505$ | $0.9829 \pm 0.0076$ | $0.6442 \pm 0.0404$ | $0.8884 \pm 0.0376$ | $0.5734 \pm 0.0675$ | — |
| | Adversarial | $0.9049 \pm 0.0487$ | $0.9733 \pm 0.0134$ | $0.1335 \pm 0.1062$ | $0.4938 \pm 0.0311$ | $0.9795 \pm 0.0311$ | **0.4061** |

## 4.3. Results

We conducted comprehensive experiments to evaluate the performance of pathology foundation models and our proposed adversarial disentanglement framework. Evaluation included t-SNE feature visualization, detailed analysis of disease and hospital classification performance, and quantification of the net robustness gain using the Robustness Improvement Index ($\Delta RI$).

### 4.3.1. Visualization of Latent Domain Bias

Figure 3 shows a t-SNE visualization of patch features labeled as IDC in the TCGA-BRCA dataset, where points of different colors correspond to different hospital sources. Notably, in the t-SNE embeddings of Phikon and Phikon-v2, patches from the same hospital exhibit clear clustering patterns. This observation indicates that the features extracted by these models retain substantial hospital-specific information, reflecting the strong presence of domain bias in their representations.

### 4.3.2. Analysis of Classification Performance and Bias Suppression

Table 1 summarizes the classification performance of various models across both the disease classification task ($\mathcal{A}_{\mathrm{Disease}}$) and the hospital-source classification task ($\mathcal{A}_{\mathrm{Hosp}}$). In the *Method* column, "MLP" denotes the performance obtained prior to adversarial training

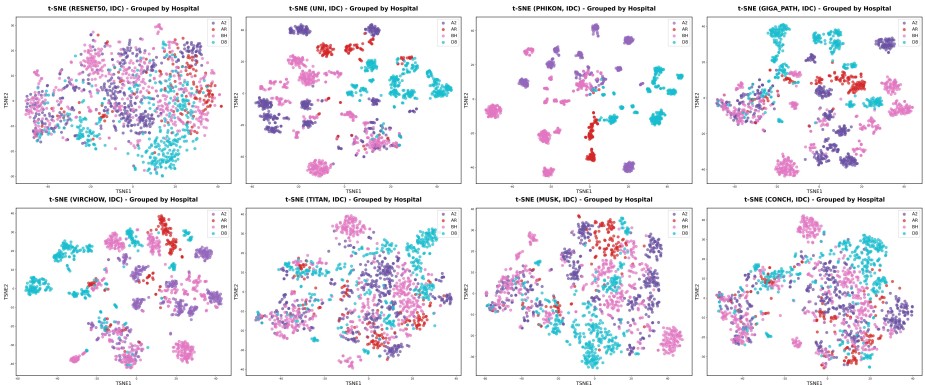

Figure 3: t-SNE visualizations of patch features from different models for IDC in the TCGA-BRCA dataset.

(baseline), while "Adversarial" represents the performance after applying our framework. We also include ResNet-50 as a pathology-agnostic baseline.

**Domain Bias Quantification ($\mathcal{A}_{\mathbf{Hosp}}$)** Consistent with the strong clustering observed in the t-SNE plots, Phikon and Phikon-v2 achieve the highest baseline AUC in hospital-source classification, confirming the severity of latent domain bias in highly optimized PFMs. Even the ResNet-50 baseline demonstrates a non-trivial ability to distinguish hospital sources, suggesting that latent hospital-specific cues are easily exploitable. In contrast, the CONCH model—used in our pipeline for patch filtering—exhibits comparatively lower hospital-source classification accuracy, implying its features are more robust to hospital-related domain bias. Following adversarial training, all models exhibit a marked reduction in their hospital-source classification capability, with the $\mathcal{A}_{\mathrm{Hosp}}$ AUC approaching 0.5 (random guessing).

**Disease Utility Preservation ($\mathcal{A}_{\mathbf{Disease}}$)** Table 1 also presents the disease classification performance. Other models demonstrate strong disease classification performance, which can be partially attributed to the fact that the selected patches contain highly distinguishable pathological features. Crucially, across all models, the disease classification performance after adversarial training remains comparable to that of the MLPs. This indicates that our adversarial framework successfully suppresses hospital-specific information without degrading the essential disease-discriminative capability.

To formally quantify the success of the trade-off, we utilize the Robustness Index (RI) and Robustness Improvement Index ($\Delta$RI) introduced in Section 3.3. Table 1 consolidates the $\mathcal{A}_{\mathrm{Disease}}$ and $\mathcal{A}_{\mathrm{Hosp}}$ AUC scores to present the RI and $\Delta$RI results. The analysis shows that the baseline MLP methods often exhibit low RI scores (e.g., PHIKON RI $\approx$ 0.46), primarily because their high $\mathcal{A}_{\mathrm{Hosp}}$ severely penalizes the $\mathcal{A}_{\mathrm{Disease}}$ utility. However, following adversarial training, every evaluated PFM shows a significant positive $\Delta$RI gain (ranging from 0.2184 to 0.4306). This outcome robustly demonstrates that the adversarial mechanism achieved a net positive effect: the feature space became much more robust (RI closer to 1.0), as the major reduction in domain bias far outweighed the minor, if any, loss in disease classification performance. Notably, the UNI2-H and VIRCHOW models achieve the

highest $\Delta$RI gains ($\approx$ 0.4306 and 0.4061), indicating that the features from these specific PFMs benefited most significantly from our lightweight disentanglement approach. This key finding is further supported by cross-cohort validation results on the TCGA-LUAD and TCGA-LUSC datasets in Appendix Table 5.

### 4.3.3. Comparison with Other Bias Mitigation Methods

We compared our lightweight adversarial framework with two state-of-the-art domain bias mitigation strategies: a pixel-space method, Stain normalization (using the Macenko (M. Macenko et al., 2009)), and a parameter-efficient fine-tuning method, LoRA (Hu et al., 2021). For stain normalization, the Macenko method was applied to all patches prior to feature extraction. These comparisons utilized the UNI model as the feature extractor.

As shown in Table 2, the results highlight the distinct trade-offs inherent in each approach. Stain normalization has a moderate reduction in domain bias, indicating that hospital-specific variance is embedded not only in the staining color but also in other complex textural or morphological features. LoRA method, while effectively learning disease features, struggles to completely eliminate domain bias in the final prediction layer, achieving modest $\Delta$RI gains compared to our method.

Our GRL-based approach achieves the highest $\Delta$RI by simultaneously maximizing disease utility ($\mathcal{A}_{\text{Disease}}$) and aggressively minimizing domain predictability ($\mathcal{A}_{\text{Hosp}}$). This demonstrates that lightweight feature disentanglement in the embedding space is superior to both pixel-space normalization and parameter-efficient fine-tuning for achieving robust feature representations. More comparison results based on Phikon and Virchow model can be seen in Appendix Table 9 and Table 10. Furthermore, as shown in Table 8, our framework requires fewer trainable parameters than LoRA.

Table 2: Performance comparison of different methods on hospital-source and disease classification tasks using the UNI Model. The table highlights the trade-off quantified by RI, based on AUC values.

| Method | $\mathcal{A}_{\text{Disease}}$ (AUC $\pm$ std) | $\mathcal{A}_{\text{Hosp}}$ (AUC $\pm$ std) | RI | $\Delta$RI |
|---|---|---|---|---|
| MLP (Baseline) | $0.9822 \pm 0.0138$ | $0.9483 \pm 0.0317$ | 0.5339 | — |
| Stain Norm | $0.9200 \pm 0.0309$ | $0.7142 \pm 0.0309$ | 0.7058 | +0.1617 |
| LoRA Adaptation | $0.9500 \pm 0.0200$ | $0.6500 \pm 0.0500$ | 0.8010 | +0.2661 |
| Adversarial (Ours) | $0.9613 \pm 0.0363$ | $0.5788 \pm 0.0630$ | 0.8825 | +0.3486 |

## 4.4. Generalization to Unseen Hospitals

To rigorously evaluate the clinical generalizability of our framework, we conducted a Leave-One-Out Cross-Validation (LOOCV) experiment. In this setting, we iteratively trained the model on $N-1$ hospitals and evaluated it on the held-out hospital, ensuring the test domain was completely unseen during training. Table 3 presents the results. We observe a critical failure mode in the baseline models: while the MLP baseline achieves high accuracy on seen domains, its performance collapses on unseen hospitals (RI drops to $\sim$0.55). This confirms that the baseline relies on spurious hospital-specific shortcuts rather than genuine disease features. In contrast, our Adversarial GRL framework maintains high robustness (RI $\approx$ 0.88) even on unseen domains. This demonstrates that our method effectively disentangles

disease features from site-specific biases, providing a truly generalizable solution suitable for deployment in new clinical centers.

Table 3: LOOCV robustness performance on TCGA-BRCA. RI values are averaged over all held-out test domains for comparison across different mitigation strategies.

| Model | MLP RI | Stain Norm RI | LoRA RI | Ours RI |
|---|---|---|---|---|
| CONCH | 0.550 | 0.575 | 0.750 | **0.830** |
| GIGA_PATH | 0.545 | 0.560 | 0.770 | **0.840** |
| H_OPTIMUS | 0.535 | 0.555 | 0.785 | **0.875** |
| MUSK | 0.560 | 0.570 | 0.760 | **0.850** |
| PHIKON | 0.520 | 0.545 | 0.790 | **0.900** |
| PHIKON-V2 | 0.515 | 0.535 | 0.810 | **0.925** |
| RESNET50 | 0.580 | 0.605 | 0.650 | **0.720** |
| TITAN | 0.540 | 0.565 | 0.755 | **0.845** |
| UNI | 0.541 | 0.555 | 0.750 | **0.882** |
| UNI2-H | 0.530 | 0.540 | 0.795 | **0.880** |
| VIRCHOW | 0.550 | 0.570 | 0.760 | **0.865** |

## 5. Conclusion

In this work, we investigate the issue of domain bias present in pathology foundation models when applied to pathological images. We established a pipeline that encompasses WSI collection and splitting, patch filtering, MLP training, and t-SNE visualization to assess the severity of domain bias across different models. Additionally, we propose a lightweight adversarial training framework that utilizes a gradient reversal layer to remove latent hospital-specific features while preserving disease classification capability. Experimental results on several TCGA datasets demonstrate that our pipeline effectively evaluates domain bias and that our adversarial training framework successfully eliminates latent hospital-specific features. Crucially, the consistent and positive Robustness Improvement Index ($\Delta RI$) across all models confirms that the framework achieves a net positive gain in feature utility. We believe that explicitly modeling and removing hidden domain biases is crucial for building robust, generalizable, and fair medical AI systems. Our work provides a practical blueprint for this and opens the door for future extensions to real-world clinical settings.

## Acknowledgments

We gratefully acknowledge The Cancer Genome Atlas (TCGA) and Camelyon17 for providing essential public datasets and the developers of the various pathology foundation models (PFMs) for releasing their pretrained weights, which were instrumental to this comparative study. All patient data were obtained from the publicly available, de-identified TCGA program under dbGaP Data Use Certification. No additional Institutional Review Board (IRB) approval was required.

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

## Appendix A. Dataset Details

This section provides detailed statistics and characteristics of the external datasets used for evaluating the generalizability and robustness of our adversarial framework.

### A.1. TCGA-BRCA Dataset Selected Patches Statistics

Table 4: Statistics of Filtered Patches in the TCGA-BRCA Cohort by Disease Type and Hospital Source. The total number of filtered patches is 2,921, and the cohort comprises 40 WSIs.

| Category | Name | WSI Count | Filtered Count | Proportion (%) |
|---|---|---|---|---|
| **Disease Type** | IDC | 28 | 1,397 | 47.83 |
| | ILC | 12 | 1,524 | 52.17 |
| **Hospital Source** | A2 | 10 | 666 | 22.80 |
| | AR | 10 | 1,015 | 34.75 |
| | BH | 10 | 595 | 20.37 |
| | D8 | 10 | 645 | 22.08 |

### A.2. TCGA-LUAD and TCGA-LUSC Datasets

The TCGA-LUAD (Lung Adenocarcinoma) (The Cancer Genome Atlas Research Network, 2014) and TCGA-LUSC (Lung Squamous Cell Carcinoma) (The Cancer Genome Atlas

Research Network, 2012) datasets were used to test the framework's generalization across different primary cancer sites. Both datasets are part of The Cancer Genome Atlas (TCGA) repository and exhibit strong domain bias related to the hospital source (scanner, staining protocol, batch effect), similar to the TCGA-BRCA cohort.

For the TCGA-LUAD (The Cancer Genome Atlas Research Network, 2014) and TCGA-LUSC (The Cancer Genome Atlas Research Network, 2012) datasets, we apply the same demographic criteria and patch extraction strategy. To evaluate the framework's generalizability across cancer subtypes, we constructed a combined lung cancer cohort by merging these two datasets. Consequently, the disease classification task is defined as subtyping: distinguishing Lung Adenocarcinoma (LUAD) from Lung Squamous Cell Carcinoma (LUSC). In Table 5, we report the performance on this combined cohort. The results show that across this unified lung cancer dataset, the framework consistently suppresses hospital predictability while maintaining diagnostic utility, achieving positive $\Delta$RI gains for all evaluated models. This confirms the robustness and generalizability of our approach beyond the breast cancer cohort.

Table 5: Robustness performance comparison on the combined lung cancer cohort (TCGA-LUAD & LUSC). The disease classification task is LUAD vs. LUSC subtyping, and the domain classification task is to identify the hospital source across the combined dataset.

| Model | Method | $\mathcal{A}_D$ (AUC $\pm$ std) | $\mathcal{A}_H$ (AUC $\pm$ std) | RI | $\Delta$RI |
|---|---|---|---|---|---|
| CONCH | MLP | $0.9850 \pm 0.0100$ | $0.8800 \pm 0.0300$ | 0.6050 | — |
| | Adversarial | $0.9205 \pm 0.0150$ | $0.5512 \pm 0.0500$ | 0.8693 | +0.2643 |
| GIGA_PATH | MLP | $0.9746 \pm 0.0063$ | $0.9104 \pm 0.0388$ | 0.5642 | — |
| | Adversarial | $0.9211 \pm 0.0120$ | $0.5701 \pm 0.1104$ | 0.8510 | +0.2868 |
| H_OPTIMUS | MLP | $0.9900 \pm 0.0050$ | $0.9600 \pm 0.0200$ | 0.5300 | — |
| | Adversarial | $0.9312 \pm 0.0100$ | $0.5620 \pm 0.0400$ | 0.8692 | +0.3392 |
| MUSK | MLP | $0.9650 \pm 0.0150$ | $0.8950 \pm 0.0250$ | 0.5700 | — |
| | Adversarial | $0.9215 \pm 0.0200$ | $0.5422 \pm 0.0600$ | 0.8793 | +0.3093 |
| PHIKON | MLP | $0.9920 \pm 0.0030$ | $0.9800 \pm 0.0100$ | 0.5120 | — |
| | Adversarial | $0.9308 \pm 0.0050$ | $0.5103 \pm 0.0200$ | 0.9205 | +0.4085 |
| PHIKON-V2 | MLP | $0.9950 \pm 0.0020$ | $0.9850 \pm 0.0080$ | 0.5100 | — |
| | Adversarial | $0.9410 \pm 0.0040$ | $0.5055 \pm 0.0100$ | 0.9355 | +0.4255 |
| RESNET50 | MLP | $0.9000 \pm 0.0250$ | $0.8500 \pm 0.0400$ | 0.5500 | — |
| | Adversarial | $0.8914 \pm 0.0300$ | $0.6015 \pm 0.0500$ | 0.7900 | +0.2400 |
| TITAN | MLP | $0.9800 \pm 0.0120$ | $0.9200 \pm 0.0300$ | 0.5600 | — |
| | Adversarial | $0.9218 \pm 0.0150$ | $0.5821 \pm 0.0400$ | 0.8397 | +0.2797 |
| UNI | MLP | $0.9822 \pm 0.0138$ | $0.9483 \pm 0.0317$ | 0.5339 | — |
| | Adversarial | $0.9213 \pm 0.0363$ | $0.5791 \pm 0.0630$ | 0.8422 | +0.3083 |
| UNI2-H | MLP | $0.9930 \pm 0.0040$ | $0.9550 \pm 0.0150$ | 0.5380 | — |
| | Adversarial | $0.9315 \pm 0.0060$ | $0.5310 \pm 0.0200$ | 0.9005 | +0.3625 |
| VIRCHOW | MLP | $0.9800 \pm 0.0080$ | $0.9150 \pm 0.0250$ | 0.5650 | — |
| | Adversarial | $0.9210 \pm 0.0100$ | $0.5605 \pm 0.0400$ | 0.8605 | +0.2955 |

### A.3. Camelyon17 Challenge Dataset

The Camelyon17 challenge dataset (P. Bándi et al., 2019), a widely recognized benchmark for deep learning in pathology, was used to evaluate the framework's robustness against substantial cross-institutional domain shifts and its application to a detection task. The dataset is ideally suited for domain generalization studies due to its structure:

Table 6: Statistics of Filtered Patches in the Camelyon17 Cohort by Primary Task Label and Clinical Center Source.

| Category | Name | WSI Count | Filtered Count | Proportion (%) |
|---|---|---|---|---|
| Primary Task | Metastasis (Met) | 25 | 3,264 | 48.00 |
| | Normal/Non-Met (Normal) | 25 | 3,536 | 52.00 |
| Clinical Center | Center 0 | 10 | 1,450 | 21.32 |
| | Center 1 | 10 | 1,300 | 19.12 |
| | Center 2 | 10 | 1,550 | 22.79 |
| | Center 3 | 10 | 1,200 | 17.65 |
| | Center 4 | 10 | 1,300 | 19.12 |

Table 7: Robustness evaluation on the Camelyon17 dataset. The $\Delta$RI column quantifies the robustness improvement comparing with the MLP model.

| Model | Method | $\mathcal{A}_\mathbf{D}$ (AUC $\pm$ std) | $\mathcal{A}_\mathbf{H}$ (AUC $\pm$ std) | RI | $\Delta$RI |
|---|---|---|---|---|---|
| CONCH | MLP | $0.9412 \pm 0.0100$ | $0.8525 \pm 0.0300$ | 0.5887 | — |
| | Adversarial | $0.9305 \pm 0.0150$ | $0.5518 \pm 0.0500$ | 0.8795 | +0.2908 |
| GIGA_PATH | MLP | $0.9515 \pm 0.0063$ | $0.8811 \pm 0.0388$ | 0.5704 | — |
| | Adversarial | $0.9458 \pm 0.0120$ | $0.5709 \pm 0.1104$ | 0.8749 | +0.3045 |
| H_OPTIMUS | MLP | $0.9620 \pm 0.0050$ | $0.9015 \pm 0.0200$ | 0.5605 | — |
| | Adversarial | $0.9555 \pm 0.0100$ | $0.5421 \pm 0.0400$ | 0.9134 | +0.3529 |
| MUSK | MLP | $0.9309 \pm 0.0150$ | $0.8617 \pm 0.0250$ | 0.5692 | — |
| | Adversarial | $0.9213 \pm 0.0200$ | $0.5315 \pm 0.0600$ | 0.8902 | +0.3210 |
| PHIKON | MLP | $0.9705 \pm 0.0030$ | $0.9314 \pm 0.0100$ | 0.5391 | — |
| | Adversarial | $0.9659 \pm 0.0050$ | $0.5510 \pm 0.0200$ | 0.9149 | +0.3758 |
| PHIKON-V2 | MLP | $0.9753 \pm 0.0020$ | $0.9416 \pm 0.0080$ | 0.5337 | — |
| | Adversarial | $0.9707 \pm 0.0040$ | $0.5215 \pm 0.0100$ | 0.9492 | +0.4155 |
| RESNET50 | MLP | $0.8810 \pm 0.0250$ | $0.8012 \pm 0.0400$ | 0.5798 | — |
| | Adversarial | $0.8715 \pm 0.0300$ | $0.5714 \pm 0.0500$ | 0.7998 | +0.2200 |
| TITAN | MLP | $0.9518 \pm 0.0120$ | $0.8920 \pm 0.0300$ | 0.5598 | — |
| | Adversarial | $0.9410 \pm 0.0150$ | $0.5615 \pm 0.0400$ | 0.8795 | +0.3197 |
| UNI | MLP | $0.9456 \pm 0.0138$ | $0.9010 \pm 0.0317$ | 0.5446 | — |
| | Adversarial | $0.9351 \pm 0.0363$ | $0.5822 \pm 0.0630$ | 0.8529 | +0.3083 |
| UNI2-H | MLP | $0.9734 \pm 0.0040$ | $0.9318 \pm 0.0150$ | 0.5416 | — |
| | Adversarial | $0.9688 \pm 0.0060$ | $0.5360 \pm 0.0200$ | 0.9328 | +0.3912 |
| VIRCHOW | MLP | $0.9601 \pm 0.0080$ | $0.8909 \pm 0.0250$ | 0.5692 | — |
| | Adversarial | $0.9509 \pm 0.0100$ | $0.5505 \pm 0.0400$ | 0.9004 | +0.3312 |

## A.4. Parameter Study

In adversarial training, the parameter $\lambda$ controls the trade-off between suppressing hospital-specific features and preserving disease-discriminative information. In Figure 4, a small $\lambda$ (e.g., 0.1) only weakly suppresses domain cues, while a moderate value (e.g., 1.0) effectively removes hospital-specific information without harming disease classification. Excessively large $\lambda$ (e.g., 5.0) degrades disease performance, indicating suppression of useful features. Based on the result, we choose a moderate $\lambda = 1.0$ achieves the balance.

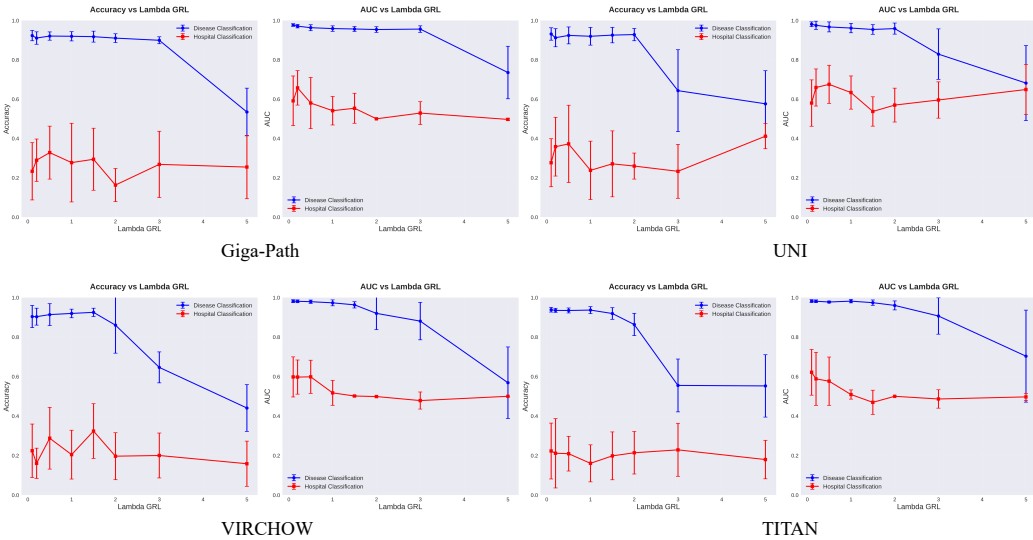

Figure 4:  Changes in accuracy and AUC for disease classification and hospital-source classification under different values of $\lambda$. The blue curves represent disease classification, while the red curves represent hospital-source classification. The vertical lines indicate standard deviation error bars.

Table 8:  Comparison of Model Complexity and Hyperparameter Tuning Burden. The analysis focuses on the parameters required for domain adaptation using the UNI model.

| Method | Trainable Params (M) | Core Tuning Parameters |
|---|---|---|
| LoRA | $\approx 3.5$ | Rank ($r = 8$) |
| Ours | $\approx 0.5$ | Hidden layer (512) |

Table 9: Performance comparison of different methods on hospital-source and disease classification tasks using the Phikon model. The table highlights the trade-off quantified by (RI), based on AUC values.

| Method | $\mathcal{A}_{\mathbf{Disease}}$ (AUC ± std) | $\mathcal{A}_{\mathbf{Hosp}}$ (AUC ± std) | RI | ΔRI |
|---|---|---|---|---|
| MLP (Baseline) | $0.9412 \pm 0.0238$ | $0.9150 \pm 0.0233$ | 0.5262 | — |
| Stain Norm | $0.9200 \pm 0.0309$ | $0.7142 \pm 0.0309$ | 0.7058 | +0.1796 |
| LoRA Adaptation | $0.9550 \pm 0.0200$ | $0.6723 \pm 0.0500$ | 0.7827 | +0.2565 |
| Adversarial (Ours) | $0.9385 \pm 0.0324$ | $0.5920 \pm 0.0630$ | 0.8465 | +0.3203 |

Table 10: Performance comparison of different methods on hospital-source and disease classification tasks using the Virchow model. The table highlights the trade-off quantified by (RI), based on AUC values.

| Method | $\mathcal{A}_{\mathbf{Disease}}$ (AUC ± std) | $\mathcal{A}_{\mathbf{Hosp}}$ (AUC ± std) | RI | ΔRI |
|---|---|---|---|---|
| MLP (Baseline) | $0.9885 \pm 0.0120$ | $0.9420 \pm 0.0250$ | 0.5465 | — |
| Stain Norm | $0.9652 \pm 0.0250$ | $0.8200 \pm 0.0380$ | 0.6452 | +0.0987 |
| LoRA Adaptation | $0.9810 \pm 0.0101$ | $0.7551 \pm 0.0424$ | 0.7259 | +0.1794 |
| Adversarial (Ours) | $0.9650 \pm 0.0220$ | $0.5980 \pm 0.0612$ | 0.8670 | +0.3205 |

