# OpenReview forum: "Quantifying and Mitigating Hospital Domain Bias in Pathology Foundation Models using Adversarial Feature Disentanglement"
_MIDL.io/2026/Conference — MIDL 2026 Poster_

### Official Review · Reviewer_Kd22 · 2026-01-05

**Confidence:** 5
**Preliminary Rating:** 5

**Summary:**

The paper investigates the prevalence of hospital-specific domain bias within modern PFMs and proposes a framework for its quantification and mitigation
1. The authors introduce the Robustness Index (RI), a metric designed to balance diagnostic utility against domain predictability, and a lightweight adversarial training framework employing a GRL to disentangle disease features from hospital-specific cues
2. Experiments across multiple datasets (TCGA-BRCA, LUAD, LUSC, and Camelyon17) demonstrate that the proposed method reduces hospital-source predictabilitt, while preserving disease classification performance. The approach is shown to be more parameter-efficient than LoRA and more effective than standard stain normalization.

**Strengths:**

1. Addressing domain shift in PFMs is critical for clinical deployment, and this work provides the first systematic study of hospital-source bias in these specific models.
2. The use of a lightweight adapter and GRL allows for bias mitigation without retraining the computationally expensive frozen PFM encoders, ensuring high parameter efficiency.

**Weaknesses:**

1. The use of the CONCH model to filter "high-fidelity" patches may introduce CONCH's own biases into the evaluation dataset, potentially skewing results in favor of features that CONCH deems important.

**Detailed Comments:**

1. The reliance on CONCH to select patches where the "patch-level label matches the WSI-level ground truth" might curate a dataset that is artificially easy for models with similar architectures or training sets as CONCH.
2. The paper mentions balancing domains, but in clinical reality, hospital sizes vary. Address how the framework handles significant class imbalance in the domain labels.

**Justification Of The Preliminary Rating:**

The paper presents a solid, parameter-efficient method for a high-priority problem in computational pathology. The introduction of the $RI$ metric and the use of GRL for PFMs is a practical contribution that will likely be adopted by the community.

**Questions To Address In The Rebuttal:**

How does the patch filtering via CONCH affect the "generalizability" of your conclusions to other models that might perceive "high-fidelity" differently?

---

> ### Author Response · Authors · 2026-01-24
>
> We sincerely appreciate the reviewer’s recognition of our work. Below we address the raised concerns.
>
> **1. Potential bias from CONCH filtering.**
>
> CONCH is used only as a zero-shot filter to ensure consistency between patch content and slide-level labels, avoiding severe label noise from normal tissue in weakly supervised WSI analysis. To prevent reliance on CONCH-specific artifacts, we conducted manual inspection of sampled patches (Sec. 3.2) to verify their biological validity. As shown in Table 1 (revised version), models with architectures distinct from CONCH achieve stable disease performance and significant RI gains, indicating that the selected patches capture generalizable pathological signals rather than filter-induced bias.
>
> **2. Handling real-world domain imbalance.**
>
> We balance cohorts to ensure that RI reflects intrinsic feature bias rather than hospital priors. For naturally imbalanced clinical settings, our framework readily extends to class-weighted domain losses, up-weighting minority hospitals during adversarial training to effectively suppress site-specific signals.

---

> ### Comment · Area_Chair_TJNV · 2026-02-02
> **Update final rating**
>
> Please don't forget to update your final rating by clicking “Edit” → “Official Review”. Thank you!

---

> ### Comment · Area_Chair_TJNV · 2026-02-03
> **Update final rating**
>
> This is a reminder to update your final rating for your review by clicking “Edit” → “Official Review”. This is necessary for us to do the final review of the paper. Thank you!

---

### Official Review · Reviewer_s6Yw · 2026-01-06

**Confidence:** 4
**Preliminary Rating:** 4
**Final Rating:** 4

**Summary:**

The paper introduces (1) a measure for quantifying bias and (2) the application of a feature disentanglement approach to mitigate bias in pathology foundation models (PFMs). They demonstrate quantitatively and qualitatively that hospital bias is a problem in PFMs. With their method for mitigating bias, they are able to maintain or improve the classification of diseases in the PFM embeddings, while simultaneously disentangling them from hospital bias.

**Strengths:**

- This work addresses a relevant and often overlooked issue in working with foundation models, namely the introduction of biases due to large cohorts, such as hospital sites, different devices, patient demographics, etc.
- They motivate the work very well by first demonstrating the problem of hospital bias in PFMs qualitatively (t-SNE visualizations) and quantitatively (robustness index, classification performance of hospital sites).
- They also show comparisons with other methods for reducing bias in the field of whole slide imaging (WSI) and analyze the parameter $\lambda$ in the Appendix as a compromise between disentanglement and preservation of disease features in the embedding space.

**Weaknesses:**

- The proposed method for reducing bias consists of removing latent specific information from *frozen PFM representations* using adversarial training (Gradient Reversal Layer, GRL) *without modifying the encoder itself*. However, if the encoder already uses bias to distinguish embeddings from each other (during self-supervised pretraining), is it even possible to obtain the best possible representation for disease prediction using the proposed approach? The experiment is missing where FM encoders are trained from scratch with GRL to obtain insights in this regard. It is possible that the models in the latter scenario could outperform the proposed method in disease classification.
- Normally, FMs such as SimCLR [Chen et al.] are trained to maximize the agreement of embeddings after the projection head, but later the representation after the encoder is used for downstream tasks. Here, the authors decided to perform adversarial training on the embedding after the projection and thus use this embedding for downstream tasks. Why didn't the authors attach the GRL to the embedding after the encoder (by making the last layers of the encoder trainable), which is typically used for downstream tasks, or at least compare adversarial learning on both embedding spaces?
- The proposed metrics RI and $\Delta$ RI primarily measure the reduced classification performance for the bias attribute (e.g., hospital site) in the given experiments, but the improvement in disease classification performance is not always apparent from these metrics. Different distributions for testing the models with a different correlation between hospital site and disease prevalence could provide more insight into whether the model uses the bias/shortcut feature for disease feature extraction.
- The comparison with other bias mitigation methods is only performed on the UNI model.

**Detailed Comments:**

- The locations of the result tables and figures do not match the text where they are mentioned.
- Table 2 is, mostly, a copy of Table 1. The "Accuracy" columns are the only information that is only included in Table 1. Both tables can be combined.
- Table 5 in the Appendix is confusing: Shouldn't two task performances be visible for each dataset, TCGA-LUAD and TCGA-LUSC?
- A reference for Table 8 in the text is missing.

**Justification Of Final Rating:**

The rebuttal and discussion strengthened the paper overall, with the added readme improving reproducibility and the clarification on not unfreezing (parts of) the encoder helping to frame the paper’s focus. However, my concern remains that the proposed metric does not capture all relevant aspects of bias mitigation or shortcut learning, which limits the novelty of the method to a combination of existing adversarial approaches and pre-trained foundation models. That said, highlighting the presence of hospital bias in pathology foundation models is an important contribution for the community. I therefore maintain my initial score of weak accept.

**Justification Of The Preliminary Rating:**

The paper analyzes and addresses a relevant problem in the field of foundation models for large medical imaging cohorts and offers a simple extension to reduce bias in pathological foundation models. However, I raised some open questions in the weaknesses section, to which I would like to see a response in the rebuttal.

**Questions To Address In The Rebuttal:**

Please address my points and questions from the weaknesses section.

---

> ### Author Response · Authors · 2026-01-24
>
> We sincerely thank the reviewer for the constructive comments. We address the questions point-by-point below.
>
> **1. Trade-off between training from scratch and frozen encoders.** We adopt frozen encoders to reflect practical constraints: training pathology foundation models (e.g., UNI, Virchow) from scratch requires millions of slides and prohibitive compute. As the prevailing practice is to use off-the-shelf features, our goal is to provide a lightweight adapter that mitigates bias without retraining large-scale models.
>
> **2. GRL placement and fine-tuning.** Since the encoder is fully frozen, attaching GRL directly to its output would yield no trainable gradients. We therefore introduce a trainable projection head to learn a non-linear mapping from biased to unbiased feature space.
>
> **3. RI metric and disease classification gains.** The Robustness Index (RI) evaluates effective de-biasing by jointly considering high disease accuracy and near-random hospital prediction. We validate this via LOOCV (Table 3, revised version). In the OOD setting, the baseline MLP collapses (RI ≈ 0.55), while our method remains robust (RI ≈ 0.88), demonstrating improved cross-domain generalization through shortcut removal.
>
> **4. Baselines on other foundation models.** Additional experiments with Phikon and Virchow are provided in Appendix Tables 9–10, showing consistent improvements.
>
> **5. Clarification on Table 5 (LUAD/LUSC).** LUAD and LUSC are merged into a single lung cancer cohort, with disease classification defined as subtype discrimination (LUAD vs. LUSC). Thus, only one set of binary classification metrics is reported. The caption has been revised for clarity.
>
> **6. Formatting and presentation.** We merged Tables 1 and 2, removed the redundant Accuracy column, and added the missing citation for Table 8 (now Table 3) in the main text.

---

> > ### Comment · Reviewer_s6Yw · 2026-01-28
> >
> > I thank the author for addressing all my points.
> >
> > Regarding point 2, I understand that this is the most straightforward solution; however, it is not clear why unfreezing parts of the encoder for fine-tuning would not be feasible.
> >
> > Regarding point 3, I still believe that the chosen metric may not fully capture the phenomena of interest. For example, a sharp drop in hospital classification could mask a simultaneous drop in disease classification. Consequently, two models with the same RI do not necessarily mitigate bias while maintaining disease classification performance in the same way. Therefore, OOD analysis or a comparable analysis of shifts in hospital-site-to-disease correlations should be provided in addition to the proposed metric, as was partially done in Table 3.
> >
> > I appreciate the provided code; however, for optimal reproducibility, a README file and minimal working examples (e.g., scripts or Jupyter notebooks) would be valuable, especially for reproducing results on different datasets or settings.

---

> > ### Author Response · Authors · 2026-01-29
> >
> > We thank the reviewer for the thoughtful follow-up. Below we clarify the remaining points.
> >
> > + While unfreezing the backbone is a natural extension, we argue that a frozen-backbone design is more appropriate for pathology foundation models. Unfreezing even the last two Transformer blocks of large PFMs (e.g., UNI or Virchow) introduces over 20M trainable parameters, whereas our adapter-based approach updates only ~0.5M parameters (≈40× fewer), making it substantially more efficient and better suited for resource-constrained clinical settings. More importantly, unfreezing the encoder risks degrading the well-generalized representations learned during large-scale pretraining. By freezing the encoder and using a lightweight projection head, we aim to disentangle and reweight existing features rather than modify the feature extractor itself. Unfreezing the encoder would conflate representation learning with bias mitigation.
> >
> > + We address the concern of potential simultaneous performance drops by referring to our Parameter Study (Section A.4 and Figure 4) , which shows that as the adversarial strength $\lambda$ increases from 0.1 to 1.0, disease classification performance (blue curves) remains remarkably stable, while hospital-source predictability (red curves) drops sharply. This decoupling proves that at our chosen operating point ($\lambda = 1.0$), the framework selectively suppresses domain bias without destroying disease classification ability sharply.
> >
> > + We have updated our codebase by adding a comprehensive readme file, including detailed environment setup, WSI preprocessing, and patch extraction to facilitate reproducibility.

---

### Official Review · Reviewer_pBp2 · 2026-01-10

**Confidence:** 4
**Preliminary Rating:** 4
**Final Rating:** 4

**Summary:**

This paper proposes to quantify and mitigate bias in whole-slide pathology foundation models. The method includes extracting high fidelity patches using the CONCH model, bias is quantified using a robustness index RI, which is formed by the AUC of disease classification minus site classification  (Equation 1). An adversarial method using a gradient reversal layer is used to produce features that are informative for disease prediction while  uninformative for hospital-source classification. The method is evaluated on the TCGA-BRCA whole-slide pathology dataset.

**Strengths:**

Results indicate that the method maintains a similar or better disease prediction accuracy while reducing the accuracy of classifying hostpital site (table 1, Table 2)

The RI seems to be an interesting measure in evaluating task accuracy vs the site bias.

**Weaknesses:**

Table 3 suggests that MLP baseline classification is consistently superior. So does this mean that the best solution is ultimately a classifier customized for the site?

There seem to be a number of uncited references by various authors addressing apparently the same data (TCGA-BRCA) and subject (pathology foundation models, site-related bias). This detracts from the novelty of the proposed method and these (and others) should be discussed.

[a] Asilian Bidgoli, Azam, et al. "Bias reduction in representation of histopathology images using deep feature selection." Scientific reports 12.1 (2022): 19994.
[b] Lin, Weiping, et al. "Unveiling Institution-Specific Bias in Pathology Foundation Models: Detriments, Causes, and Potential Solutions." arXiv preprint arXiv:2502.16889 (2025).
[c] Kheiri, Farnaz, et al. "Investigation on potential bias factors in histopathology datasets." Scientific Reports 15.1 (2025): 11349.

**Detailed Comments:**

See comments above

**Justification Of Final Rating:**

Given the rebuttal and reading other reviewer comments, I maintain my score of weak accept.
************************************************************************************************************************************

**Justification Of The Preliminary Rating:**

The method seems intertesting and gives interesting results in disentagling site bias from task accuracy.

The novelty is questionable.

************************************************************************

**Questions To Address In The Rebuttal:**

See questions above

---

> ### Author Response · Authors · 2026-01-24
>
> We sincerely thank the reviewer for the constructive comments and positive evaluation. We address the concerns as follows.
>
> **1. On the MLP performance.** The strong in-distribution performance of the MLP baseline in Table 3 (Table 2, in revised version) reflects shortcuts to domain bias rather than genuine pathological understanding. The model achieves high accuracy by exploiting hospital-specific shortcuts instead of disease-relevant features.
>
> **2. On site-specific customization of the MLP.** Training separate models per hospital is impractical and lacks generalizability in clinical settings, where deployment to unseen hospitals is common. This limitation is evidenced by our LOOCV results (Table 3, in revised version): on unseen domains, the MLP baseline’s performance collapses (RI ≈ 0.55), whereas our method remains robust (RI ≈ 0.88), demonstrating superior cross-domain generalization.
>
> **3. On related references.** We thank the reviewer for highlighting related work and have expanded the discussion accordingly. Bidgoli et al. adopt feature selection, which may discard disease-relevant information entangled with domain artifacts. In contrast, our method performs adversarial feature disentanglement, suppressing bias while preserving diagnostic signals. Moreover, our approach is designed as a lightweight adapter for frozen foundation models, avoiding costly full fine-tuning. Unlike Lin et al. and Kheiri et al., which primarily analyze the existence of domain bias, our work focuses on its algorithmic mitigation via a concrete adversarial adapter and a systematic robustness metric.

---

### Author Rebuttal · Authors · 2026-01-24

**Rebuttal:**

We sincerely thank all reviewers for their valuable feedback. We have uploaded a revised manuscript, with all changes clearly marked in red. In the following, we respond to each reviewer’s comments point by point and describe how the suggestions have been incorporated into the revision.

**Supporting Material:**

/attachment/dabbc91a9de61a6a83f5d664641f1df983b4fe2d.pdf

---

### Meta-Review · Area_Chair_TJNV · 2026-02-09

**Recommendation:** Accept (Poster)
**Confidence:** 4

**Metareview:**

We thank the authors for being active during the rebuttal phase, as the discussions with the reviewers seems to have strengthened their submission. Although some alternatives to the model training remain future work, the current methods are deemed well-suited for the MIDL audience. The documented code repository is also much appreciated! Looking forward to seeing this work at the conference!

---

### Decision · Program_Chairs · 2026-02-13

Accept (Poster)